# Selinexor and Other Selective Inhibitors of Nuclear Export (SINEs)—A Novel Approach to Target Hematologic Malignancies and Solid Tumors

Kajetan Karaszewski * and Wiesław Wiktor Jędrzejczak

Department of Hematology, Transplantation, and Internal Medicine, Medical University of Warsaw, Żwirki i Wigury 61, 02-091 Warsaw, Poland; wieslaw.jedrzejczak@wum.edu.pl
* Correspondence: kajetan.karaszewski@gmail.com

**Abstract:** Exportin 1 (XPO1) is a crucial molecule of nucleocytoplasmic transport. Among others, it exports molecules important for oncogenesis from the nucleus to the cytoplasm. The expression of XPO1 is increased in numerous malignancies, which contributes to the abnormal localization of tumor suppressor proteins in the cytoplasm and subsequent cell cycle dysregulation. Selective inhibitors of nuclear export (SINEs) are novel anticancer agents that target XPO1, arrest tumor suppressor proteins in the nucleus, and induce apoptosis in cancer cells. Selinexor, a first-in-class SINE, has already been approved for the treatment of relapsed/refractory multiple myeloma and relapsed/refractory diffuse large B cell lymphoma not otherwise specified. It has also been proven effective in relapsed/refractory and previously untreated acute myeloid leukemia patients. In addition, numerous studies have yielded promising results in other malignancies of the hematopoietic system and solid tumors. However, future clinical use of selinexor and other SINEs may be hampered by their significant toxicity.

**Keywords:** selinexor; selective inhibitor of nuclear export; multiple myeloma; diffuse large B cell lymphoma; acute myeloid leukemia



## 1. Introduction

The transport of molecules between the nucleus and the cytoplasm is a crucial process underlying the biology of eukaryotic cells. The nuclear envelope successfully separates DNA replication and RNA transcription in the nucleus from protein synthesis and their further modification in the cytoplasm. Small molecules can diffuse passively through the nuclear pore complex (NPC) [1,2]. However, most macromolecules require nucleocytoplasmic transport factors in order to reach the other side of the nuclear envelope [3]. These factors belong predominantly to the family of karyopherin-β. They are subdivided into exportins, which transport cargo to the cytosolic compartment, and importins, which are able to take cargo to the nuclear compartment [1,3,4].

Nuclear export requires the conversion of RanGTP to RanGDP, and phosphate to provide energy for the process. Therefore, in order to reach the cytoplasm through NPC, an exportin, RanGTP, and a cargo must create a complex. After the process, RanGDP is transported back to the nucleus via NPC, in the presence of nuclear transport factor 2 (NTF2). The exportin itself shuttles back from the cytoplasm into the nucleus via NPC, where it is ready to begin another cycle. Afterward, RanGDP is phosphorylated in the nucleus to RanGTP by GTP, and therefore, further export is possible. A high concentration of RanGTP in the nucleus is granted by the regulator of chromosome condensation 1 (RCC1), an element essential to maintain the Ran cycle [1–3,5]. The Ran cycle is shown in Figure 1.

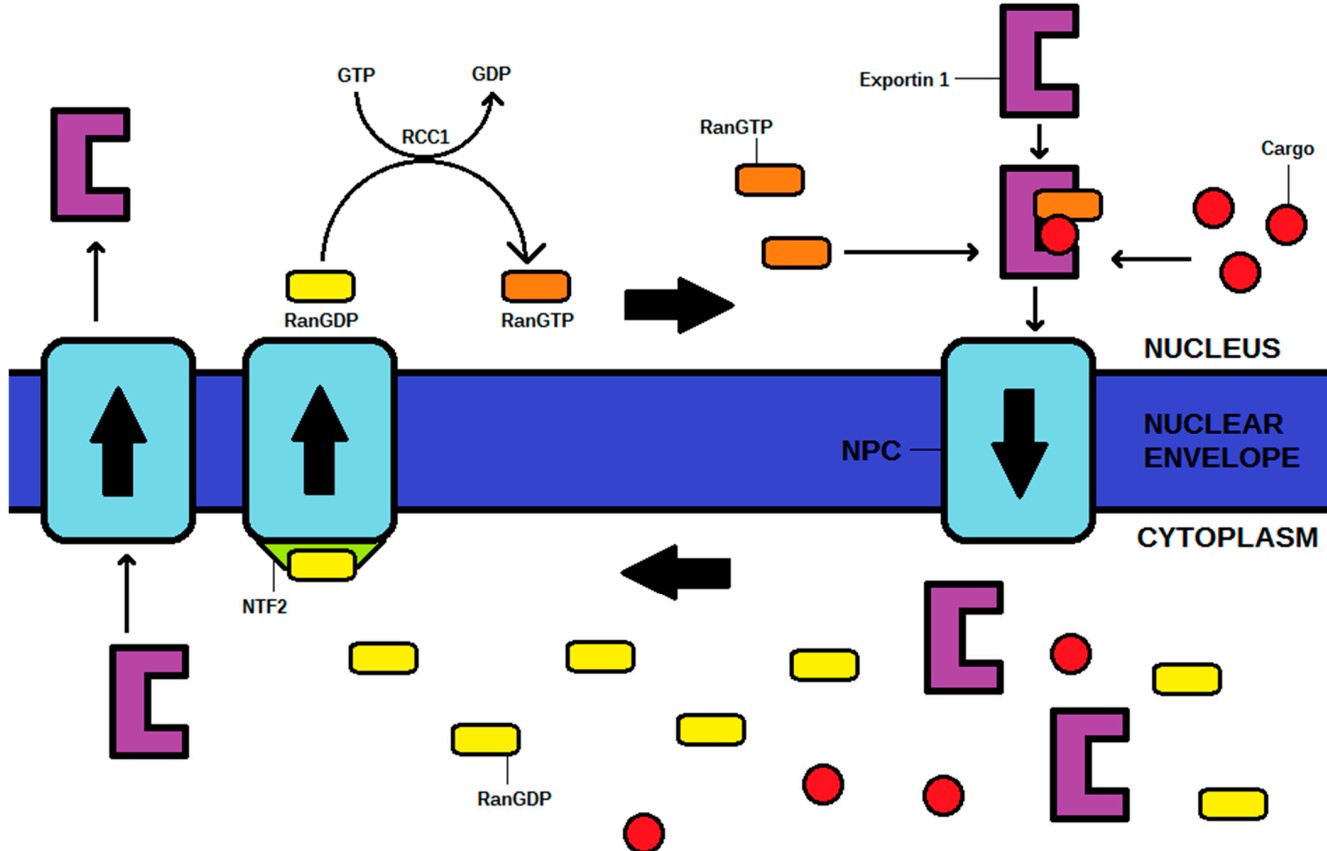

**Figure 1.** The mechanism of the Ran cycle and exportin (exportin 1 as the example) transporting a cargo from the nucleus to cytoplasm in exportin 1-RanGTP-cargo complex. NPC—nuclear pore complex, NTF2—nuclear transport factor 2, RCC1—regulator of chromosome condensation 1.

Chromosome maintenance protein 1 (CRM1), also known as exportin 1 (XPO1), is the first protein confirmed to provide nuclear export [6] and the most crucial exportin known to date [3]. XPO1 transports over 200 different cargoes, such as transcriptional factors, translational factors, or kinases [7,8]. The comprehensive molecular mechanism of the exportin 1–RanGTP cargo complex creation has not been fully explained yet. Notably, transported cargoes must contain leucine-rich nuclear export signals (NES), while XPO1 in an unbound state has a ring-like hydrophobic structure which contains a NES-binding domain [4]. Moreover, XPO1 contains a loop in one of HEAT repeats, which regulates cargo binding through an allosteric mechanism [3]. The architecture of XPO1 was first modeled in 2004 based on X-ray crystallography, homology modeling, and finally, electron microscopy [9].

Some of the transported proteins are protooncogene products or tumor suppressor proteins, and therefore, XPO1 might impact oncogenesis [7]. Furthermore, the overexpression or increased activity of XPO1 is a common phenomenon in cancer cells, leading to nuclear deprivation of critical suppressor and regulatory proteins, such as p53, p21, IκB, RB, p27 [4,8,10]. As a result, these proteins are no more effective in counteracting genetic aberrations after being exported. Therefore, the cell cycle becomes dysregulated, leading to abnormal proliferation [8]. For instance, such a phenomenon was confirmed in breast cancer with *BRCA1* mutation, where the altered localization of the *BRCA1* protein product in the cytosolic compartment promotes metastasis [11].

Furthermore, XPO1 exports miRNAs that regulate cellular quiescence, which is the reversible state of proliferative arrest [12]. However, this is only an alternative pathway, as exportin 5 seems to play a more significant role in miRNA transport than exportin 1 [13]. The role of miRNA transport in oncogenesis is still poorly understood, although it is claimed

that genetic abnormalities, such as deletions, might downregulate miRNA expression in some cancers. It leads to the dysregulation of quiescence/proliferation checkpoints and creates an opportunity for growth factors to increase proliferation [14]. Such a phenomenon occurs in chronic lymphocytic leukemia (CLL) with 13q14 deletions (more than half of CLL cases), where neoplastic cells are deprived of miRNA15 and miRNA16 [15].

Exportin 1 might also induce resistance to chemotherapy and to targeted cancer therapies by preventing the agents from achieving proper concentration in the nucleus [4]. For example, XPO1 is responsible for resistance to imatinib, ibrutinib, or cisplatin [5,10]. Overall, inducing pharmacotherapy resistance and escaping the cell cycle as a result of exporting tumor suppressor proteins both contribute to the poor prognosis of cancers with increased XPO1 expression [16].

The action of XPO1 might be inactivated by an antibiotic leptomycin B, which attaches covalently to the cysteine-528 residue of the NES-binding domain [4,5,17]. Although successful in vitro and in vivo, this drug has never been used by clinicians due to its unacceptable toxicity, and subsequent recommendations to cease further clinical trials [5,8,18]. XPO1 inhibition has also been achieved using other molecules such as antibiotics (anguinomycins, ratjadones), natural substances (goniothalamin, valtrate, curcumin), and most importantly, using selinexor and other selective XPO1 inhibitors such as eltanexor [4]. These drugs, known as selective inhibitors of nuclear export (SINEs), have the same mechanism of action as leptomycin B, targeting the cysteine-528 of XPO1. SINE molecules contain hydrophobic trifluoromethyl groups buried deeply in the NES-binding domain of XPO1. However, the most significant part of their structures is the triazole scaffold, which forms covalent yet reversible bindings with the cysteine-528 of XPO1 [1]. The cellular mechanism of action is shown in Figure 2. The chemical structures of SINEs are shown in Figure 3.

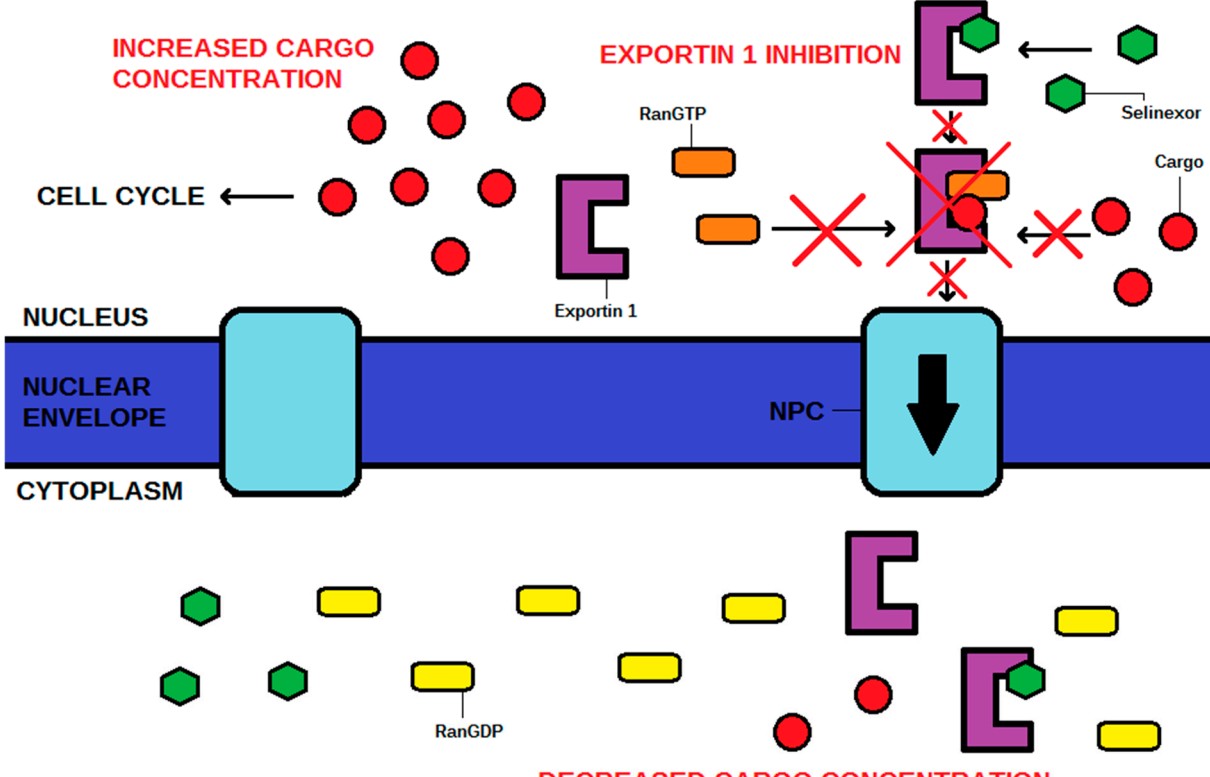

**Figure 2.** The mechanism of action of selinexor, a selective inhibitor of nuclear export (SINE). The cargo is a tumor suppressor protein that rebuilds the cell cycle. NPC—nuclear pore complex.

**Figure 3.** The chemical structures of selective inhibitors of nuclear export (SINEs).

This review aims to summarize up-to-date clinical applications and preclinical findings, and to discuss the potential future role of this category of drugs. It is necessary to search for promising agents and new cancer treatment methods, as described malignancies of the hematopoietic system and solid tumors remain a challenge for clinicians, despite improvements in overall survival. In particular, the options for heavily pretreated patients with relapsed or refractory disease are still limited, as mentioned in the newest reviews and guidelines for multiple myeloma [19,20], diffuse large B-cell lymphoma [21,22], and acute myeloid leukemia [23,24].

## 2. Selinexor in R/R MM

Selinexor (KPT-330, brand name—Xpovio) is the first-in-class oral SINE targeted at exportin 1. The drug has been approved for the treatment of relapsed/refractory multiple myeloma (R/R MM) [25–27]. It was registered by the FDA in July 2019 in combination with dexamethasone for patients who have undergone at least four prior therapies and are refractory to at least two proteasome inhibitors (PIs), at least two immunomodulatory agents (IMIDs), and anti-CD38 monoclonal antibodies (MABs) [25,27]. Although these three classes of drugs are crucial in MM treatment, the majority of patients become refractory to them at some point of the therapy ("triple refractory MM") [26]. Some patients undergo even more therapies (for instance, autologous stem cell transplantation) before selinexor is administered, with a median of seven lines of prior treatment revealed by some studies [28,29]. Management algorithms suggest considering selinexor in heavily pretreated MM patients as an alternative to BCL-2 inhibitor venetoclax, immunotherapy or CAR-T cell therapies [30]. In December 2020, selinexor was approved by the FDA in combination with bortezomib and dexamethasone for patients who have undergone one prior therapy only [31].

As SINE, selinexor affects cancer cells by trapping tumor suppressor proteins and oncoprotein mRNA in the nucleus and rebalancing the cell cycle [16]. Additionally, selinexor increases the expression of glucocorticoid receptors and inhibits the mTOR pathway synergistically with dexamethasone. Hence, an accelerated registration pathway was used for

combination of selinexor with dexamethasone [29,32]. Moreover, selinexor inhibits DNA repair mechanisms in cancer cells and sensitizes these cells to DNA-damaging agents [33]. Therefore, clinical trials assessing the effectiveness of selinexor + dexamethasone and another additional agent (bortezomib [34,35], carfilizomib [36,37], pomalidomide [38] or doxorubicin [39]) have also been performed. Hence, it is still possible that the indications could be widened, and selinexor could be used in earlier stages of MM in other combinations, before the development of refractoriness to many other agents. Preclinical studies in this field are also ongoing [40]. The combination of venetoclax and selinexor was proven effective in R/R MM with translocation t(11;14) [41]. Moreover, selinexor was confirmed to overcome hypoxia-induced drug resistance to bortezomib, which supports the use of the currently approved combination: selinexor + bortezomib + dexamethasone [42].

Selinexor in combination with dexamethasone produced a 26% overall response rate (ORR) in R/R myeloma. The median duration of remission was 4.4 months, the median overall survival (OS) was 8.6 months, and the median progression-free survival (PFS) was 3.7 months [28,43]. Other studies yielded similar results (the ORR of 21%, the median duration of remission of 5 months) [29]. For the combination of selinexor + carfilizomib + dexamethasone, the overall response rate (ORR) was 38% and the median OS was 22.7 months [36,44]. The combination of selinexor + bortezomib + dexamethasone produced the ORR of 63% in patients with a median of three prior therapies [44,45]. However, these studies did not include a control group. The only phase III clinical trial assessed the efficacy of selinexor in combination with bortezomib and dexamethasone (SVd) versus bortezomib and dexamethasone (Vd) [34,46]. The median PFS was significantly longer (13.9 months vs. 9.5 months, $p = 0.0075$), and the ORR was significantly higher in SVd group (76% vs. 62%, $p = 0.0012$). The results were even more promising in terms of ORR, if only subgroups with high-risk cytogenetics were compared (ORR: 77% vs. 56%, $p = 0.0008$). However, no significant differences in terms of PFS have been observed in this subgroup. Moreover, no significant difference in OS between SVd and Vd groups has been found [34]. Considering all collected data, selinexor is still beneficial for patients with R/R MM, regardless of cytogenetic risk [47]. Interestingly, the latest cytogenetic analyses revealed that sensitivity to selinexor is strongly correlated with the expression of ABCC4 in MM cells, which implies the usefulness of ABCC4 as a predictive biomarker [48].

According to numerous studies, selinexor was not well tolerated by the patients. Due to adverse reactions, treatment was discontinued in 27% of cases. Over 50% of patients required dose reductions or dose delays, and fatal adverse effects occurred in 9% of cases [26]. Thrombocytopenia, anemia, leukopenia, neutropenia; hyponatremia; dyspnea, and upper respiratory infections occurred frequently. Less serious reactions were nausea, vomiting, diarrhea, decreased weight, and fatigue [26,38]. The occurrence of ocular adverse events was estimated at 20% [49], while neurological adverse events occurred in 25% of cases [50]. Due to the unsafe profile of selinexor, dedicated recommendations for dealing with adverse reactions have been prepared [51].

## 3. Selinexor in R/R DLBCL NOS

Selinexor was approved by FDA in June 2020 for the treatment of R/R diffuse large B cell lymphoma (DLBCL) not otherwise specified (NOS) after at least two lines of systemic therapy. As the third (or further) line of treatment, selinexor might be considered alternatively to CAR-T cell therapies and antibody-drug conjugates (ADCs) (polatuzumab vedotin, loncastuximab tesirine) [50].

Selinexor in monotherapy produced an ORR of 29%, where 38% of responses lasted at least 6 months. The median duration of remission of 9.3 months was observed, while the expected survival for R/R DLBCL patients is generally less than 6 months [50,52]. In another study, the ORR was estimated at 28%, the median PFS at 3.6 months, and the median OS at 9.1 months [53,54]. Notably, similar ORRs were achieved regardless of DLBCL subtype (germinal center B cell-like, non-germinal center B cell-like), but ORRs were lower

for double-hit and triple-hit subtypes [55,56]. Patients with other non-Hodgkin lymphoma (NHL) types (follicular lymphoma, mantle cell lymphoma, and Richter transformation) also achieved an ORR of approximately 30% [50,55]. This would imply the possibility of creating new drug combinations in the near future. Moreover, in NHL, selinexor was proven to enhance the effectiveness of standard therapy R-CHOP (rituximab, cyclophosphamide, doxorubicin, vincristine, prednisone) [57].

XPO1 overexpression was confirmed to worsen the prognosis of DLBCL patients with unfavorable cytogenetics (*BCL-2* overexpression, double-hit, triple-hit) [56]. Moreover, targeting XPO1 with selinexor is only effective in the absence of the *TP53* mutation. Otherwise, resistance is induced [58]. Therefore, selinexor does not seem to be beneficial for patients with unfavorable cytogenetic changes, but this phenomenon requires confirmation on larger groups of patients in further studies. Comprehensive genomic profiling in DLBCL revealed the recurrent mutation E571K of XPO1 [57]. Its role in pathogenesis and potential responsibility for inducing resistance is still unclear. However, it might be another target for future novel drugs.

Statistical analysis revealed that the adverse events of monotherapy with selinexor did not have a clinically meaningful negative impact on patients' quality of life, despite the adverse events of grade 3 or 4 experienced in over 80% of cases [59]. Therefore, despite the unsafe profile of selinexor, adverse reactions during therapy should be considered manageable. It is worth noticing that the treatment response and stable disease groups were associated with significantly higher quality of life than the group of patients who experienced progressive disease. This would imply that the incidence and severity of adverse events during therapy had an impact on dose reductions or treatment discontinuation, and therefore affected the eventual outcome.

## 4. Selinexor in Other Hematologic Malignancies

Research on selinexor as an anticancer agent is not limited to its current indications. Its efficacy in acute myeloid leukemia (AML) has been assessed in numerous preclinical and clinical trials. Moreover, compassionate use in some cases has also been reported [60].

Selinexor with a CLAG (cladribine, cytarabine, filgrastim) regimen in R/R AML patients produced a complete remission (CR) rate of 45% [61]. In another study, the combination of selinexor and decitabine in R/R AML produced an ORR of 30%, a CR rate of 25%, a median OS of 5.9 months, and a median PFS of 5.9 months [62]. With selinexor as monotherapy, the ORR was 14%, the median OS among responders was 9.7 months, and the median PFS among responders was 5.1 months [63].

In previously untreated AML patients, selinexor added to standard therapy (daunorubicin + cytarabine: '3+7') produced a significantly higher CR rate in comparison with standard therapy alone (80% vs. 59%, *p* = 0.018) [64]. Moreover, selinexor + '3+7' was proven to be a safe regimen, and it produced a median OS of 10.3 months in previously untreated AML patients [65]. Maintenance therapy with selinexor after allogeneic stem cell transplantation in high-risk AML patients was also proven safe and effective [66]. A meta-analysis of the drug's efficacy and safety in AML treatment is in progress [67]. Therefore, new approval for an XPO1 inhibitor in AML is probable in the near future. Moreover, an ORR of 26% was achieved in patients with myelodysplastic syndrome (MDS) or oligoblastic AML, which implies the possibility of using an XPO1 inhibitor at even earlier stages of the disease [64,68].

Moreover, in mouse AML models, selinexor was proven to synergize with topoisomerase inhibitors, increase sensitivity to idarubicin, and reduce DNA damage repair [69]. A synergistic anticancer effect with azacitidine was confirmed in AML cell lines [70].

Selinexor in combination with DICE (dexamethasone, ifosfamide, carboplatin, etoposide) produced an ORR of 82% and a 1-year survival of 67% in patients with R/R peripheral T cell lymphoma (PTCL) or natural killer/T cell lymphoma (NKTL) [71]. It is an alternative for other targeted agents in PTCL treatment (PI3K inhibitors, monoclonal antibodies, ADC—brentuximab vedotin) [72].

Selinexor was proven to synergize with ibrutinib, to increase OS in mouse CLL models, and to be effective in ibrutinib-resistant CLL in vitro [73]. The efficacy of selinexor in monotherapy was confirmed in CLL cells in vitro [74].

Selinexor combined with imatinib selectively targets chronic myeloid leukemia (CML) stem cells, implying the possibility of successfully eliminating residual disease in patients resistant to imatinib alone [75].

Examples of clinical studies involving selinexor in malignancies of the hematopoietic system are shown in Table 1.

**Table 1.** Examples of clinical studies involving selinexor in malignancies of the hematopoietic system (alphabetical order). Abbreviations: AML—acute myeloid leukemia, CR—complete remission, DICE—dexamethasone, ifosfamide, carboplatin, etoposide, DLBCL—diffuse large B cell lymphoma, MDS—myelodysplastic syndrome, MM—multiple myeloma, NHL—non-Hodgkin lymphoma, ORR—overall response rate, PFS—progression-free survival, R-CHOP—rituximab, cyclophosphamide, doxorubicin, vincristine, prednisone, '3+7'—daunorubicin + cytarabine therapy.

| Number of Reference | Type of Malignancy | Type of Study | Year of Publication | Outcomes/ Conclusions |
|---|---|---|---|---|
| [61] | AML | phase I clinical study | 2020 | CR = 45% in combination with cladribine, cytarabine and filgrastim |
| [62] | AML | phase I clinical study | 2020 | CR = 25%, PFS = 5.9 months in combination with decitabine |
| [63] | AML | phase I clinical study | 2017 | ORR = 14% and safe profile in monotherapy |
| [64] | AML (previously untreated) | phase II clinical study | 2022 | significantly higher CR rate in combination with standard therapy '3+7' than '3+7' alone |
| [65] | AML (previously untreated) | phase I clinical study | 2019 | OS = 10.6 months and safe profile in combination with '3+7' |
| [68] | AML (+MDS) | phase II clinical study | 2020 | ORR = 26% in monotherapy |
| [50] | DLBCL | phase II clinical study | 2021 | ORR = 29% in monotherapy |
| [54] | DLBCL | phase II clinical study | 2020 | ORR = 28%, PFS = 3.6 months in monotherapy |
| [55] | DLBCL (and other NHL) | phase I clinical study | 2017 | ORR = 31% and safe profile in monotherapy |
| [57] | DLBCL (and other NHL) | phase I clinical study | 2021 | synergistic effect with R-CHOP therapy, safe profile |
| [28] | MM | phase I clinical study | 2019 | ORR = 26%, OS = 8.6 months in combination with dexametasone |
| [36] | MM | phase I clinical study | 2019 | ORR = 38%, OS = 22.7 months in combination with dexamethasone and carfilizomib |
| [45] | MM | phase I clinical study | 2018 | ORR = 63% in combination with dexamethasone and bortezomib |
| [34] | MM | phase III clinical study | 2020 | significantly higher ORR rate and PFS in combination with dexamethasone and bortezomib vs. dexamethasone and bortezomib alone |
| [39] | MM | phase I/II clinical study | 2017 | ORR = 15% in combination with dexamethasone and doxorubicin |
| [38] | MM | phase I clinical study | 2019 | ORR = 31% and PFS = 12.2 months in combination with dexamethasone and pomalidomide |
| [71] | T-cell lymphomas | phase I clinical study | 2021 | ORR = 82% and 1-year survival of 67% in combination with DICE |

## 5. Selinexor in Solid Tumors

Selinexor has been approved for malignancies of the hematopoietic system, and current research is mainly focused on its use in these diseases. However, it has also been considered for the therapy of solid tumors [76].

Selinexor has demonstrated anticancer activity in advanced gynecological malignancies [77,78]. In a phase I clinical study of selinexor with carboplatin and paclitaxel (CP) in patients with ovarian or endometrial cancer, the ORR was 57%, and the regimen was proven safe [77]. Furthermore, in the phase II clinical study of patients with recurrent ovarian, endometrial or cervical cancer, single-agent selinexor produced the CR rates of 30%, 35%, and 24%, respectively, and a median OS of 7.3 months, 7.0 months, and 5.0 months, respectively. Moreover, in this study, selinexor was safe for and tolerated by the patients. The majority of adverse events were mild (grade 1 or 2), reversible and fully manageable with supportive care [78].

Selinexor reduced angiogenesis, tumor growth, and the incidence of metastases and increased the OS in preclinical models of prostate cancer [79]. However, clinical activity could not be fully assessed, as the phase II clinical study of patients with refractory, castration-resistant metastatic prostate cancer was terminated before completion due to unacceptable toxicity of selinexor in combination with abiraterone and enzalutamide [80].

Selinexor was proven to decrease the concentration of hypoxia-inducible factor 1 (HIF-1) and, subsequently, to decrease radioresistance in human osteosarcoma cell lines [81]. Notably, selinexor successfully inhibited the growth of xenografts derived from other sarcoma types (liposarcoma, leiomyosarcoma, rhabdomyosarcoma, gastrointestinal stromal tumor, undifferentiated sarcomas) [82]. However, despite potential activity in numerous preclinical studies of sarcoma cell lines [83,84], the phase I clinical study did not yield promising results. Single-agent selinexor did not produce an objective response (by RECIST) in any of the evaluated patients [85].

Additionally, despite significant tumor reduction, no partial or complete responses were observed in the phase II clinical study of single-agent selinexor in patients with recurrent or metastatic salivary gland tumors [86].

There are numerous in vitro/in vivo preclinical studies that have assessed the efficacy of selinexor in these and other solid tumors. The examples are presented in Table 2. However, despite promising results and a further improvement in our understanding of the underlying molecular background, actual clinical prospects are still limited, and the safety profile is questionable.

**Table 2.** Examples of preclinical and clinical studies involving selinexor in solid tumors (alphabetical order). Abbreviations: CR—complete remission, OS—overall survival, PFS—progression-free survival.

| Number of Reference | Type of Solid Tumor | Type of Study | Year of Publication | Outcomes/Conclusions |
|---|---|---|---|---|
| [87] | anaplastic thyroid carcinoma | in vitro/in vivo preclinical study | 2017 | synergistic effect with daunorubicin |
| [88] | bladder malignancies | in vitro preclinical study | 2018 | decreased tumor growth |
| [89] | breast cancer (triple negative) | in vitro preclinical study | 2017 | antitumor activity in monotherapy |
| [90] | breast cancer (triple negative) | in vitro preclinical study | 2021 | antitumor activity in combination with olaparib |
| [91] | chordoma | in vivo preclinical study | 2022 | tumor growth inhibition in 78–92% in combination with abemaciclib |
| [92] | colorectal cancer | in vitro/in vivo preclinical study | 2017 | synergistic effect with bortezomib |
| [93] | colorectal cancer | in vitro/in vivo preclinical study | 2016 | increased efficacy of anticancer radiation |
| [94] | gastric cancer | in vitro preclinical study | 2018 | synergistic effect with irinotekan |
| [95] | glioblastoma | in vitro/in vivo preclinical study | 2015 | decreased tumor growth, increased OS |

**Table 2.** *Cont.*

| Number of Reference | Type of Solid Tumor | Type of Study | Year of Publication | Outcomes/Conclusions |
|---|---|---|---|---|
| [96] | glioblastoma | in vitro/in vivo preclinical study | 2018 | increased efficacy of anticancer radiation |
| [97] | head and neck squamous cell carcinoma | in vitro preclinical study | 2018 | reversion of anthracycline resistance |
| [98] | hepatoma, osteosarcoma | in vitro preclinical study | 2021 | reversion of radioresistance |
| [83] | liposarcoma | in vitro/in vivo preclinical study | 2017 | decreased tumor growth |
| [84] | liposarcoma | review article | 2022 | - |
| [99] | lung adenocarcinoma | in vitro preclinical study | 2021 | decreased tumor growth |
| [100] | neuroblastoma | in vitro preclinical study | 2021 | synergistic effect with bortezomib |
| [77] | ovarian cancer, endometrial cancer | phase I clinical study | 2020 | safety and good tolerance of selinexor + carboplatin therapy |
| [78] | ovarian cancer, endometrial cancer, cervical cancer | phase II clinical study | 2019 | CR rates of 30%, 35% and 24% respectively, median OS of 7.3 months, 7.0 months and 5.0 months respectively |
| [79] | prostate cancer | in vitro/in vivo preclinical study | 2014 | decreased tumor growth, increased OS |
| [80] | prostate cancer | phase II clinical study | 2018 | anticancer activity, yet poor tolerability in combination with abiraterone and enzalutamide |
| [101] | renal cell carcinoma | in vitro/in vivo preclinical study | 2014 | anticancer activity similar to sunitinib |
| [86] | salivary gland tumor | phase II clinical study | 2022 | tumor reduction in 61%, median PFS of 4.9 months |
| [85] | sarcoma | phase I clinical study | 2016 | partial responses, no CRs, safety and good tolerability in monotherapy |
| [82] | sarcoma | in vitro/in vivo preclinical study | 2016 | anticancer activity in different sarcoma subtypes |
| [81] | sarcoma | review article | 2021 | - |

## 6. Other SINEs

Eltanexor (KPT-8602) is a second-generation SINE proven to inhibit XPO1 in malignancies of the hematopoietic system in preclinical models in vitro and in vivo. In AML and DLBCL patient-derived xenografts, eltanexor exerts a significant synergistic effect when co-administered with a BCL-2 inhibitor (venetoclax) [102]. Moreover, the drug demonstrates potent activity in both B cell and T cell acute lymphoblastic leukemia (ALL) models in vivo [103], and synergizes with dexamethasone in these malignancies in vitro [104].

In the phase I clinical trial, oral high-dose eltanexor produced an ORR of 40%, a median PFS of 4.5 months, and a median OS of 17.8 months in patients with R/R MM [105]. These results are promising, but they require a comparative study with selinexor in order to determine actual usefulness in the clinic. In patients with high-risk MDS that is refractory to hypomethylating agents (azacitidine), eltanexor produced an ORR of 53%, a CR of 47% and a median OS of 9.9 months [106]. Eltanexor has also been investigated in preclinical models of solid tumors, with promising results in glioblastoma [107] and castration-resistant prostate cancer [108]. The severity and incidence of adverse reactions associated with eltanexor were lower than for selinexor in R/R MM patients. The majority of events were cytopenias and gastrointestinal abnormalities, and rarely, mild neurological abnormalities. Patient withdrawal due to adverse events was only 8% [105].

KPT-185 is a SINE that has demonstrated anticancer activity in mantle cell lymphoma (MCL) in vitro [109] and in MM in vitro [110]. However, the efficacy was lower in MCL cells with high expression of *TP53* [111]. In in vitro and in vivo preclinical models of AML, KPT-185 induced the apoptosis and downregulation of the *FLT3* oncogene [112]. This activity has also been shown in non-small-cell lung cancer in vitro and in vivo [113].

KPT-276 is a SINE of similar properties to KPT-185. Both agents have demonstrated co-activity in MCL [114] and NHL [115] cell lines. Moreover, KPT-276 was proven effective in MM in vitro and in vivo, which contributed to the initiation of the phase 1 clinical trial in this indication [116].

Verdinexor (KPT-335) has demonstrated anticancer activity in preclinical models of esophageal cancer [117] and neuroblastoma [118] in vitro and in vivo. However, the research on this SINE is mainly focused on antiviral activity against the influenza A virus [119] and RSV [120]. Interestingly, according to the most recent studies, selinexor has demonstrated antiviral activity against Merkel cell carcinoma virus [121,122]. Apparently, the potential of this category of drugs is not limited to neoplastic diseases.

Other examples of using SINEs in preclinical models of malignancies of the hematopoietic system are felezonexor (SL-401) in CML, AML, MM, and Hodgkin lymphoma [123]; KPT-251 in AML [124]; and CBS9106 in MM [125]. Although all these drugs significantly decreased the proliferation of cancer cells in vitro, no clinical trials have been initiated for many years, and these drugs are far from clinical application. The most probable candidate for a second-generation SINE in cancer therapy is eltanexor. Its efficacy and safe profile have already been proven by the phase I clinical studies in the years 2021–2022. Hopefully, a safer alternative for selinexor will find its place in clinics in the near future. The chemical structures of selinexor and other SINEs are shown in Figure 3. Examples of studies involving other SINEs in neoplastic diseases are shown in Table 3.

**Table 3.** Examples of studies involving selective inhibitors of nuclear export other than selinexor in malignancies of the hematopoietic system and solid tumors (alphabetical order). Abbreviations: ALL—acute lymphoblastic leukemia, AML—acute myeloid leukemia, DLBCL—diffuse large B cell lymphoma, MCL—mantle cell lymphoma, MM—multiple myeloma, NHL—non-Hodgkin lymphoma, NSCLC—non-small-cell lung carcinoma, ORR—overall response rate, OS—overall survival.

| Number of Reference | Selective Inhibitor of Nuclear Export | Type of Study | Year of Publication | Outcomes/Conclusions |
|---|---|---|---|---|
| [125] | CBS9106 | in vitro preclinical study | 2011 | decreased proliferation of MM cells |
| [102] | eltanexor | in vitro/in vivo preclinical study | 2020 | synergistic effect with venetoclax in AML and DLBCL models |
| [103] | eltanexor | in vitro/in vivo preclinical study | 2017 | decreased proliferation of ALL cells |
| [104] | eltanexor | in vitro/in vivo preclinical study | 2020 | synergistic effect with dexamethasone in ALL models |
| [105] | eltanexor | phase I clinical study | 2021 | OS = 17.8 months, ORR = 40% in combination with dexamethason in MM |
| [106] | eltanexor | phase I clinical study | 2022 | ORR = 53% in high-risk myelodysplastic syndrome |
| [107] | eltanexor | in vitro preclinical study | 2022 | anticancer effect in glioblastoma cells |
| [108] | eltanexor | in vitro preclinical study | 2021 | synergistic effect with PARP inhibitors in prostate cancer cells |
| [123] | felezonexor (SL-401) | in vitro preclinical study | 2016 | anticancer effect in numerous hematologic malignancies |
| [109] | KPT-185 | in vitro preclinical study | 2012 | anticancer effect in MCL |
| [110] | KPT-185 | in vitro preclinical study | 2011 | anticancer effect in MM |
| [111] | KPT-185 | in vitro preclinical study | 2014 | anticancer effect via p53-dependent mechanism in MCL |
| [112] | KPT-185 | in vitro/in vivo preclinical study | 2012 | decreased proliferation of AML cells, increased OS |
| [113] | KPT-185 | in vitro/in vivo preclinical study | 2014 | anticancer effect in NSCLC |
| [124] | KPT-251 | in vitro/in vivo preclinical study | 2013 | anticancer effect in AML |

**Table 3.** *Cont.*

| Number of Reference | Selective Inhibitor of Nuclear Export | Type of Study | Year of Publication | Outcomes/Conclusions |
|---|---|---|---|---|
| [114] | KPT-276 (+ KPT-185) | in vitro/in vivo preclinical study | 2013 | anticancer effect in MCL |
| [115] | KPT-276 (+ KPT-185) | in vitro/in vivo preclinical study | 2014 | decreased proliferation of NHL |
| [116] | KPT-276 | in vitro preclinical study | 2013 | decreased proliferation of MM |
| [117] | verdinexor | in vitro/in vivo preclinical study | 2022 | decreased proliferation and migration of esophageal cancer |
| [118] | verdinexor | in vitro/in vivo preclinical study | 2021 | anticancer effect in neuroblastoma |

## 7. Summary and Future Directions

Selinexor is a promising agent already registered for the second or further line therapy of R/R MM [31] and in the third or further line therapy of DLBCL NOS [50]. The drug has already been and is still being assessed in numerous clinical and preclinical studies. Significant achievements have been demonstrated in R/R AML [64,65], which might contribute to the approval of selinexor for this indication in the near future. Although research has mainly focused on malignancies of the hematopoietic system [126], using XPO1 as a target may also be useful in solid tumors (Table 2). Moreover, due to numerous novel SINEs with promising anticancer activity, it is possible that selinexor will not be the only agent with this mechanism of action in the clinic. Eltanexor, whose efficacy has already been documented in a phase I clinical trial for R/R MM, is a possible candidate for a registered second-generation SINE in the longer term [105]. A timeline of the crucial facts and events in the history of SINEs is shown in Figure 4.

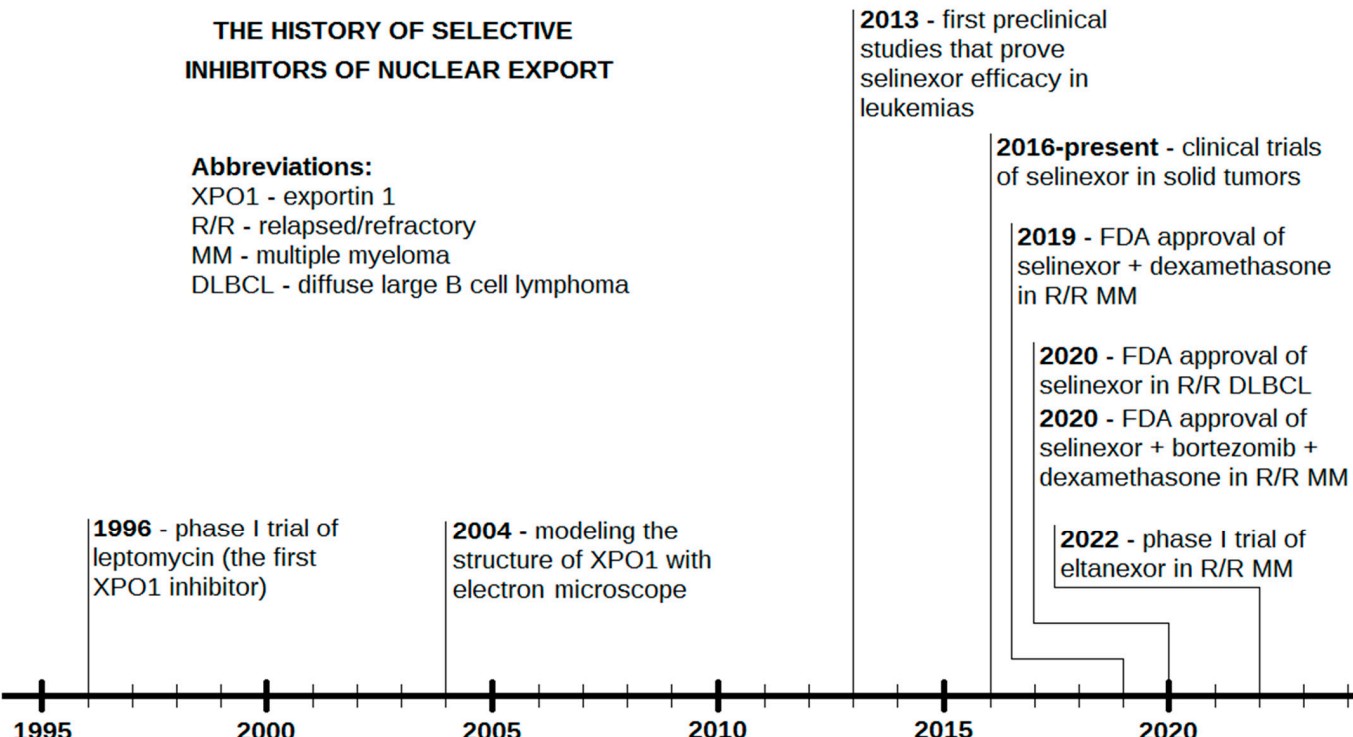

**Figure 4.** The history of selective inhibitors of nuclear export (SINEs).

On the other hand, the unsafe toxicity profile of selinexor might be a potential barrier to its clinical application. The ratio and wide spectrum of adverse events in R/R MM patients led to dedicated guidelines based on independent clinical trials [51]. In DLBCL patients, the impact on quality of life was not that explicit [59]. Moreover, unacceptable toxicity was the reason for the termination of the phase II clinical trial in patients with castration-resistant

metastatic prostate cancer [80]. On the other hand, a phase II clinical trial in patients with advanced gynecological malignancies yielded more promising results [78]. According to the only clinical trial, eltanexor had a safer toxicity profile than selinexor [105]. Other SINEs have not been investigated yet, but hopefully they will show similarities to eltanexor, leading to another SINE being applied in clinics. This would be a significant event, as the options for patients with R/R malignancies of the hematopoietic system and solid tumors are still limited.

However, even the approval and successful clinical application of next-generation SINEs would probably not constitute a breakthrough, due to the heterogeneity of neoplasms. The wide spectrum of genetic abnormalities and different protein expression profiles has a decisive impact on patients' response to treatment and prognosis in DL-BCL [127] or AML [128], as examples. Therefore, especially in relapsed/refractory diseases, personalized therapy is necessary to maximize clinical benefits. However, the more unique options are available, the more we can offer to patients, and this is the purpose of novel, promising categories of drugs such as SINEs. Considering the recent success of selinexor in different malignancies and current attempts to optimize new candidates, this category of drugs is likely to play a greater role in the future, if its toxicity profile and accessibility are improved [129].

XPO1 is a vulnerable target in different malignancies of the hematopoietic system, especially MM and solid tumors [116]. However, although XPO1 is crucial, it is not the only molecule that regulates nuclear export [130]. Multiple exportins have already been identified. Therefore, targeting one of them might not fully exploit the potential of anticancer activity. Examples of these exportins include Cse1, Pse1, Kap123, Sxm1 and Mtr10 [130]. There are also potential prospects of finding promising nuclear import inhibitors [131], the examples of which include karyostatin 1A, importazole, ivermectine, and mifepristone. Although these molecules are widely used as biological tools to identify cargo proteins, none of them have entered clinical trials for neoplastic disorders thus far [131].

Overall, nuclear transport has fundamental importance for a variety of pathophysiological processes. Therefore, understanding the interplay of exportins and importins will be crucial for the further perfection of compounds affecting nuclear transport. Hopefully, this will result in even more success of the aforementioned category of agents in the future.

**Author Contributions:** Both authors contributed to the work and meet the ICMJE criteria of authorship. All authors have read and agreed to the published version of the manuscript.

**Funding:** This research received no external funding.

**Institutional Review Board Statement:** Not applicable.

**Informed Consent Statement:** Not applicable.

**Data Availability Statement:** Not applicable.

**Conflicts of Interest:** The authors declare no conflict of interest.

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
