# Peer review of "Selinexor and Other Selective Inhibitors of Nuclear Export (SINEs)—A Novel Approach to Target Hematologic Malignancies and Solid Tumors"

_ddc, doi:10.3390/ddc2020023_

Round 1

Reviewer 1 Report

This is a well-written review article describing anticancer agents, Selinexor and other selective inhibitors of nuclear export (SINEs). This reviewer would recommend this manuscript for publication in drugs and drug candidates, if the following points could be addressed.

1.     Chemical structures of Selinexor and other SINE inhibitors including Eltanexor and Verdinexor should be indicated. The DDC readers such as medicinal chemists may want to understand the correlation between Selinexor and other SINE inhibitors in terms of chemical aspects.

2.     In Fig. 2 model, Selinexor binds to a common RanGTP-binding site of XPO1 and thereby inhibits the interaction between RanGTP and XPO1. How does it achieve selective cargo protein export inhibition through such a mechanism? If Selinexor binds to the cargo binding site of XPO1, selective inhibition can be explained.

Author Response

We are grateful for kind comments on our manuscript. We modified it according to the advice.

  1. The chemical structures of selinexor and other SINEs have been shown in a new Figure 3.
  2. The molecular mechanism of cargo binding to XPO1 has been discussed (lines 52-57 of the revised manuscript). The mechanism of selective XPO1 inhibition with regard to crucial groups in chemical structures of SINEs has been discussed (lines 86 and 93-96 of the revised manuscript).

Reviewer 2 Report

1. It is required that authors add some recent references.

2. Aim and motivation to write this review is missing.

3. Future perspectives need to be added.

4. The authors are advised to add some figures as they are essential to attract reader's attention.

5. Selinexor demonstrated anticancer activity in advanced gynecological malignancies. Line 217 ...Add a reference.

There are grammatical errors and manuscript needs proofreading.

Author Response

We are grateful for kind comments on our manuscript. We made our best efforts to improve language quality in the limited time we had at our disposal. We modified the manuscript according to the advice.

  1. Recent references (numbers 117-131) from the years 2022 and 2023 have been added to underline that the topic is still actual for the readers.
  2. The aim of the study has been defined and motivated (lines 98-105 of the revised manuscript). The motivation includes recent guidelines and reviews (numbers 117-122).
  3. Future clinical perspectives have been discussed (lines 369-378 of the revised manuscript).
  4. There is a new Figure 3, which shows the chemical structures of selinexor and other SINEs. Moreover, we added Figure 4, which depicts the timeline of key facts and events related to XPO1 and its inhibitors. Moreover, two additional tables have been added.
  5. Proper references have been added (line 259 of the revised manuscript).

Reviewer 3 Report

The authors reviewed the therapeutic and adverse effects of selective inhibitors of nuclear export, in particular selinexor, on various types of cancer, in particular on hematological neoplasms.

While these drugs are interesting and the data are presented mainly in text form, the authors should put more effort into offering the available data in a compact, summarizing and clear format, not only listing them one after the other in the text with a reference.

Specific Points of Criticism and Suggestions for Alterations:

(1)   Line 30:  Instead of "achieve", maybe use "reach".

(2)  Figure 1:  What happens to Exportin 1 once it is in the cytoplasm? Maybe this could be included in the otherwise very informative figure.

(3)  Line 54:  Commonly the listed proteins do not "eliminate" genetic aberrations. Proposal:  "counteract".

(4)  Line 115:  The correct style of writing in cytogenetics is "t(11;14)" using a semicolon [and not „t(11,14)" using a comma].

(5)  Line 129:  PFS in months with 2 decimals should be avoided, suffice 1 decimal or even better no decimals at all. Also the percentages of ORR and of other parameters with decimals is pseudo-exact - the term is actually "percentage" (not promille/per thousand).

(6)  The significant toxicity should be mentioned in the Summary and in the Abstract. As a phase II trial was stopped due to unacceptable toxicity (lines 225-228) and as the main drug was not well tolerated with significant adverse events (lines 139-147), then this should be emphasized and the applicability of the drug should be questioned/disucssed.

(7)  Table 1:  Is there some type of order in Table 1 how the studies are listed? They are certainly not arranged alphabetically, for example according to the first letter of the tumor.  An additional column referencing the outcome and/or conclusion would be useful.

(8)  A table similar to Table 1 for the hematological diseases would be useful ? with outcomes?

(9)  A table for the other SINEs could also be included.

Egnlish is generally okay, only a few minor corrections are necessary.

Author Response

We are grateful for kind comments on our manuscript. We included new Figures and Tables which present the topic more compactly, so that the article is more attractive to the readers. We modified the manuscript according to the advice.

  1. The word “achieve” was replaced by “reach” (line 31 of the revised manuscript).
  2. Figure 1 was modified in order to clarify that unbound Exportin 1 is transported back to the nucleus via NPC after the hydrolysis of the complex. We also included this information in the main text (lines 39-40 of the revised manuscript).
  3. The word “eliminating” was replaced by “counteracting” (line 64 of the revised manuscript).
  4. The cytogenetic aberration was rewritten correctly, with a semicolon (line 139 of the revised manuscript).
  5. We analyzed the whole article regarding this suggestion, and we rewritten the durations of survivals and remissions with a precision of 0.1 months and survival and remission rates with a precision of 1%.
  6. We included additional information about the toxicity of eltanexor (lines 307-310 of the revised manuscript), selinexor in gynecological malignancies (lines 246-266 of the revised manuscript) and DLBCL (lines 202-206 of the revised manuscript). We discussed the aspect of the drugs’ unsafe profile in the Summary section (lines 357-368 of the revised manuscript). In the Abstract, we clarified that the unsafe profile might be a barrier to future clinical application of selinexor and other SINEs. We made other minor changes in the Abstract so that it is not longer than 150 words, according to the author guidelines of the journal.
  7. We used alphabetical order to present studies in Table 1 (in the revised manuscript it is Table 2). We included general outcomes/conclusions in a distinct column.
  8. We added a new table (Table 1 of the revised manuscript), which shows the examples of studies involving selinexor in malignancies of the hematopoietic system.
  9. We added the new Table 3, which shows the examples of studies involving other SINEs in neoplastic diseases.

Round 2

Reviewer 1 Report

The authors adequately addressed this reviewer's comment.